# Association between Depressed Mood and Sleep Duration among Various Ethnic Groups—The Helius Study

**DOI:** 10.3390/ijerph18137134

**Published:** 2021-07-03

**Authors:** Kenneth Anujuo, Karien Stronks, Marieke B. Snijder, Anja Lok, Girardin Jean-Louis, Charles Agyemang

**Affiliations:** 1Department of Public & Occupational Health, Amsterdam UMC, Amsterdam Public Health Research Institute, 1105 AZ Amsterdam, The Netherlands; k.stronks@amsterdamumc.nl (K.S.); m.b.snijder@amsterdamumc.nl (M.B.S.); c.o.agyemang@amsterdamumc.nl (C.A.); 2Department of Clinical Epidemiology, Biostatistics and Bioinformatics, Amsterdam UMC, Amsterdam Public Health Research Institute, 1105 AZ Amsterdam, The Netherlands; 3Department of Psychiatry, Amsterdam UMC, location AMC, University of Amsterdam, 1105 AZ Amsterdam, The Netherlands; a.lok@amsterdamumc.nl; 4Department of Population Health, Center for Healthful Behavior Change, New York University School of Medicine, New York, NY 10016, USA; girardin.jean-louis@nyulangone.org

**Keywords:** sleep duration, depression, ethnicity, minority groups, HELIUS study

## Abstract

*Background:* We examined the association between depressed mood (DM) and sleep duration in a multi-ethnic population in Amsterdam, and the extent to which DM accounts for both short and long sleep. *Methods:* Cross-sectional data using 21,072 participants (aged 18–71 years) from the HELIUS study were analyzed. Sleep duration was classified as: short, healthy, and long (<7, 7–8, and ≥9 h/night). A Patient Health Questionnaire (PHQ-9 sum score ≥10) was used to measure DM. The association between DM and sleep duration was assessed using logistic regression. The extent to which DM accounted for short and long sleep was assessed using a population attributable fraction (PAF). *Results:* DM was significantly associated with short sleep in all ethnic groups after adjustment for other covariates (OR 1.9 (1.5–2.7) in Ghanaians to 2.5 (1.9–32) in the Dutch). DM was not associated with long sleep except in the Dutch (OR 1.9; 1.3–2.8). DM partly accounted for the prevalence of short sleep with PAF ranging from 3.5% in Ghanaians to 15.5% in Turkish. For long sleep, this was 7.1% in the Dutch. Conclusions: DM was associated with short sleep in all ethnic groups, except in Dutch. If confirmed in longitudinal analyses, strategies to reduce depression may reduce the prevalence of short sleep in concerned groups.

## 1. Introduction

Depression is a complex mental disorder that may cause disability and morbidity in the general population. Depression causes more disability than any other disorder, and it is estimated to affect about 350 million people across the globe [1,2]. Major depressive disorder (MDD) is one of the most prevalent psychiatric disorders and is estimated to become the leading cause of disease burden in high-income countries by 2030 [3].

A previous longitudinal study conducted in the United States of America (USA) has shown that adolescents with symptoms of depression at baseline had higher rates of sleep problems at a 4-year follow-up than did those who had not experienced depressive symptoms at baseline [4]. This suggests that depressive symptoms may influence sleep disruption. For instance rumination due to depression may precipitate sleep problems such as insomnia and hypersomnia. Certain studies have also shown that aberrant sleep duration was associated with depressive symptoms [5,6], and yet other studies have demonstrated a bi-directional relationship between sleep duration and depression [7,8,9]. In addition, sleep problems are also part of the diagnosis of depressive disorder and therefore may also share common risk factors and biological features, and the two conditions may respond to some of the same treatment strategies [10].

This paper focuses on the association between depressive symptoms and sleep duration in a multi-ethnic population in particular. Among ethnic minority groups, there is a high prevalence of both depressive symptoms and short sleep compared to the host population, which has, for instance, been demonstrated in the Netherlands [11,12,13]. However, the extent to which depressive symptoms are associated with sleep duration across different ethnic groups has not yet been elucidated. In the current study, we examined the association between depressed mood and sleep duration among six ethnic groups. In addition, we quantified to what extent depressed mood accounted for the prevalence of short and long sleep duration in each ethnic group.

## 2. Materials and Methods

### 2.1. Study Population

The current study was based on baseline data from the HELIUS (Healthy Life in an Urban Setting) study. The aims and design of the HELIUS study have been described elsewhere [14,15]. In brief, HELIUS is a large-scale cohort study on health and health care among different ethnic groups living in Amsterdam. The study includes individuals aged 18–70 years from the six major ethnic groups in Amsterdam (African Surinamese, South-Asian Surinamese, Turkish, Moroccan, Ghanaian, and Dutch origin), and focuses on three major disease categories: cardiovascular disease, mental health, and infectious diseases. Participants were randomly sampled from the municipal registers and stratified by ethnicity. Baseline data were collected by questionnaires and a physical examination. The depression and sleep items questionnaires were presented in local languages and translated into English, and well understood by participants. The study protocols were approved by the AMC Ethical Review Board. All participants provided written informed consent.

Baseline data collection took place in 2011–2015. Data from both the questionnaires and the physical examination were available for22,165 participants. For the current analyses, participants with Javanese Surinamese origin (n = 233), other/unknown Surinamese origin (n = 267), and other/unknown origin (n = 48) were excluded from the analysis because of small sample sizes, resulting in 21,617 participants. Subsequently, individuals with missing data on sleep duration (n = 334) or depressed mood (n = 211) were excluded from the analysis. This resulted in a dataset of 21,072 participants, including 4540 Dutch, 2992 South-Asian Surinamese, 4052 African Surinamese, 2192 Ghanaians, 3494 Turks, and 3802 Moroccans.

### 2.2. Ethnicity

Ethnicity was defined according to the country of birth of the participant as well as that of parents [16]. Specifically, a participant was considered of non-Dutch ethnic origin if he/she fulfilled either of the following criteria: (1) he or she was born abroad and has at least one parent born abroad (first generation); or (2) he or she was born in the Netherlands but both his/her parents were born abroad (second generation). Of the Surinamese migrants in the Netherlands, approximately 80% were either of African or South-Asian origin. Both subgroups were classified according to the self-reported ethnic origin. For the Dutch sample, we included people who were born in the Netherlands and whose parents were born in the Netherlands.

### 2.3. Depressive Symptoms

Depressive symptoms were measured using the Patient Health Questionnaire-9 (PHQ-9) [17]. PHQ-9 determines depressive symptoms over the preceding 2 weeks. A validity study within the HELIUS study indicated that the PHQ-9 measures the same concept among ethnic groups and that there are no systematic differences in reporting between the ethnic groups [18]. PHQ-9 consists of nine items, with a response scale varying from 0 (never) to 3 (nearly every day). If one of the items was missing, the mean score of the other eight items was used to replace the missing item. If more than one item was missing, the variable was considered missing. A cut-off point for the sum score (ranging from 0 to 27) of ≥10 was used to determine depressed mood [12].

### 2.4. Sleep Duration

Participants were asked to provide information on the average number of hours they usually sleep at night. Sleep duration was assessed using the item, “How many hours do you sleep on average per night?” Short sleep was defined as having less than 7 h of sleep per night, healthy sleep was defined as having 7–8 h of sleep per night, and long sleep was defined as having 9 or more hours of sleep per night, in line with National Sleep Foundation (NSF), American Academy of Sleep Medicine (AASM), and the Sleep Research Society (SRS) recommendations [19]. Sleep duration was measured in full hour increments.

### 2.5. Other Measurements

Educational level was determined using the participant’s highest level of education obtained (either in the Netherlands or in the country of origin). Participants were categorized into those who have never been to school or had elementary schooling only (1st category), those with lower vocational schooling or lower secondary schooling (2nd category), those with intermediate vocational schooling or intermediate/higher secondary education schooling (3rd category), and those with higher vocational schooling or university (4th category). For the current analyses, the first two categories were combined because of the small numbers of these categories among the Dutch. Occupational level was defined according to the Dutch Standard Occupational classification, based on job title and description, and was categorized into the following five groups: elementary, lower, middle, higher, and scientific profession. Alcohol intake in the past 12 months (yes/no) and current smoking status (yes/no) were also obtained by the questionnaire. Participants were asked to bring their prescribed medications to the physical examination, which were then coded according to the Anatomical Therapeutic Chemical (ATC) classification [20]. When using medications coded as (H1_Psychotroop), participants were considered to use psychotropic medications.

### 2.6. Data Analysis

Characteristics of the study population, by ethnicity, were expressed as percentages or means with 95% confidence intervals. The relationship between depressed mood (independent variable) and short sleep duration (outcome variable) was analyzed per ethnic group, by the use of logistic regression with adjustments for potential confounders (age and sex) and covariates (marital status, education, occupation, alcohol use, smoking, and the use of a psychotropic substance) and were expressed as odds ratios (ORs) with 95% confidence intervals (CIs). Effect modification by ethnicity was tested by adding interaction terms (ethnic groups*depression) to these regression models including all ethnic groups. We did not stratify the analysis by gender because there was no interaction between ethnicity and gender in the association between depressed mood and sleep duration (*p* = 0.160).The contribution of depression to sleep duration was calculated using a population attributable fraction (PAF) for each ethnic group, as proposed by Williamson 2010 and Rockhill 1998 [21,22]. The PAF was interpreted as the proportional reduction in the prevalence of short or long sleep duration that would occur if, ideally, no one has depression. All analyses were performed using STATA 14.2 (Stata Corp, College Station, TX, USA).

## 3. Results

### 3.1. Characteristics of the Study Population

Table 1 shows the characteristics of the study population by ethnicity.Moroccan and Turkish participants were younger, consumed alcohol less often, had lower educational levels, and had a higher prevalence of long sleep and depressed mood, compared with the other ethnic groups. Similar to Turkish and Moroccan participants, South-Asian Surinamese also had a higher prevalence of depressed mood compared with the other ethnic groups. The prevalence of depressed mood was about two times higher in South-Asian Surinamese, and three times higher in Turks and Moroccans, compared to the Dutch. Ghanaians also had a lower educational level, more often an elementary occupation, and used psychotropic medication less often than the Dutch and other ethnic groups. South-Asian Surinamese, African Surinamese, and Ghanaian participants had a lower mean sleep duration and a higher prevalence of short sleep than the Dutch, Turks, and Moroccans. Ghanaian and Moroccan groups had the lowest percentage of smokers, whereas the Turkish group had the highest.

### 3.2. Association between Depressed Mood and Short Sleep Duration

Figure 1 shows the prevalence of short sleep duration by ethnicity and the presence of depressed mood. There was a significant interaction between the ethnic groups and depression (*p* < 0.001). The prevalence of short sleep was consistently higher in participants who had depressed moods than those who did not have depressed mood, in all ethnic groups. The observed association remained significant after adjustment for age and sex, marital status, education, occupation, alcohol use, smoking, and use of psychotropic medications, with ORs ranging from 1.9 in Ghanaian migrants to 2.5 in the Dutch (Table 2).

### 3.3. Association between Depressed Mood and Long Sleep Duration

Figure 2 shows the prevalence of long sleep duration by ethnicity and depressed mood. The prevalence of long sleep was higher in participants who had depressed mood than among those who did not have depressed mood, in all ethnic groups. However, the observed association was no longer significant after adjustment for age and sex, marital status, education, occupation, alcohol use, smoking, and use of psychotropic medication, except in the Dutch (OR 2.5; 1.3–2.8) (Table 2).

### 3.4. The Extent to Which Depressed Mood Accounted for the Prevalence of Short and Long Sleep Duration among Ethnic Groups

Using PAF, we calculated to what extent depressed mood goes hand in hand with the prevalence of short and long sleep duration in each ethnic group (Table 2). For short sleep, the PAF in Dutch was 6.7% (95% CI 4.4–9.0), suggesting that the prevalence of short sleep would be reduced by 6.7% if, ideally, no one among them had depressed mood, suggesting the short sleep is the result of a depressed mood. Similarly, among the ethnic minority groups, the PAF ranged from 3.5% in Ghanaians to 15.5% in Turkish people, respectively. For long sleep, the PAFs for the Dutch was 7.1%.

## 4. Discussion

In this study, we investigated to what extent the higher prevalence of depressed mood accounted for a higher prevalence of short sleep duration or long sleep duration in ethnic minority groups. Depressed mood was consistently associated with short sleep in all ethnic groups. Depressed mood accounted for a moderate part (15.5% and 13.9%) of the prevalence of short sleep in Turks and Moroccans, but less so in other ethnic minority groups. Depressed mood was not associated with long sleep, except in the Dutch in whom depressed mood accounted for a moderate part (7.1%) of the prevalence of long sleep. To our knowledge, this is the first study that examined the association between depressed mood and sleep duration in a multi-ethnic population.

The overlap between the prevalence of short and long sleep on the one hand, and depressed mood on the other hand, might partly reflect the impact of depressed mood on sleep problems. Patten et al. (2000) [4], in a previous population-based longitudinal study conducted in the USA, demonstrated that adolescents with symptoms of depression at baseline had an increased rate of sleep problems at the 4-year follow-up than did those who had not experienced depressive symptoms at baseline. In a Norwegian-based study, Silvertsen et al. (2014) [9] also reported that depression was cross-sectionally associated with sleep duration among adolescents. Our cross-sectional results in adults partly concur with these findings.

Potential underlying mechanisms for the association between depression and sleep duration have been proposed in a previous study, which suggests that both environmental and biological factors may be involved [23]. For instance, it has been shown that people with depression have reduced self-regulatory skills that are necessary to maintain/enforce bedtime routines, which may result in variable sleeping time at night [23,24]. In addition, the dysregulation of the hypothalamic pituitary adrenal (HPA) axis, which is associated with depression [24], may negatively impact the timing and patterns of sleep due to increased levels of cortisol during the pre-sleep period [25]. Additionally, pre-sleep worry and cognitive arousal commonly observed in adults have been shown to have similar negative impacts on the timing and patterns of sleep [26,27].Moreover, insufficient/excessive sleep, as well as dysfunctions of sleep rhythm, are likely to occur during depression and are therefore part of the symptoms of depressive disorder [28]. The sleep–wake cycle is regulated by two interacting processes: the circadian process and the homeostatic (or recovery) process. The former regulates the daily rhythms of the body and the brain; this is mainly due to the suprachiasmatic nucleus of the hypothalamus, which provides an oscillatory pattern of activity regulating fundamental mechanisms, e.g., sleep–wake activity, hormone release, and liver function [28]. The effects of these mechanisms may lead to variations/disruptions in sleep duration, which manifest as short or long sleep duration.

One important observation from our study is the larger overlap between depressed mood and short sleep in Turks and Moroccans compared to other ethnic groups. This could be attributed to the higher prevalence of depressive symptoms in Turks and Moroccans compared to the other groups, as has been widely reported in previous studies [12,18]. In addition, psychosocial stress due to migration, perceived discrimination, congested and noisy residential environments, cultural practices such as communal night sleeping (co-sleeping), beliefs and attitude towards sleep, acculturation, [13], family size, differential health care, and genetics could affect sleep duration and may vary across ethnic groups, and therefore may contribute to the observed differences among ethnic groups.

Our study results indicated that the higher prevalence of short sleep in the Ghanaian and Surinamese groups does not go hand in hand with depression, suggesting that other factors, such as work-related factors [29], may account for the increased prevalence of short sleep rather than depression. Further studies should explore other potential factors contributing to the higher prevalence of short sleep in these groups.

Another observation from our study is the low prevalence of long sleep and depressed mood in the Dutch compared to other groups (except long sleep in African Surinamese). Yet, depressed mood did account for the prevalence of long sleep in the Dutch (PAF 7.1%), which is not observed in other groups. A possible explanation for the association being absent in the other ethnic minority groups could be the disadvantaged position of the ethnic minority groups, such as job insecurity. For example, because many people rely on multiple insecure jobs, they may not get the time off to address their depression problem. The strength of our study lies in the large sample size, thereby providing us the possibility to stratify for ethnicity and compare ethnic groups. Some limitations of this study include the use of self-reported data on depressive symptoms and sleep, which may be subject to recall bias; hence, participants may have under- or over-reported depressive symptoms and sleep durations. However, in the HELIUS study, it was recently shown that the PHQ-9 measured the same underlying construct in all the ethnic groups [18], which implies that the ethnic differences in depressed mood are not caused by measurement variance of the questionnaire but reflect real differences between the ethnic groups. Previous studies document that depression and sleep are tightly intertwined [30], and that the association of depressed mood with sleep could be due to biological/genetic and environmental factors. We do not have information on genetic and environmental factors in our dataset; hence, this is an additional limitation of our study. Another limitation of our study is the use of questionnaires to assess sleep duration instead of actigraphy or polysomnography, which are recognized as the gold standard for objective sleep measures. Questionnaires are subjective and prone to recall bias; however, we do not have actigraphy or polysomnography information in our dataset.

Parenthood and the number of children are factors, which may influence depressed mood and sleep duration. We did not adjust for parenthood and the number of children as we do not have the information in the original dataset. Similarly, the number of family members that live together, cohabitation, as well as co-sleeping could influence sleep duration and depressed mood. Our dataset has no information on these variables; hence, this is an additional limitation of our study. Moreover, information on daytime sleeping/naps was lacking in the original dataset and was not included in the analyses. Future studies should investigate these factors and how they contribute to the observed differences among various ethnic groups. Although the use of cross-sectional data cannot allow us to infer a causal association between depression and short or long sleep, our findings concur with a previous longitudinal study conducted on a similar subject. However, since a bi-directional association between depression and sleep duration does exist, we cannot rule out the possibility of an effect of sleep on depression; therefore, we acknowledge our inability to unravel both mechanisms. However, the discussion of the reverse direction is beyond the scope of this paper.

## 5. Conclusions

In conclusion, the results of our study indicate that depressed mood is associated with short sleep in all ethnic groups, but not with long sleep (except in Dutch). Insofar as the short sleep is the consequence of the high prevalence of depressed mood, these findings suggest that strategies aimed to promote healthy sleep may also include management of depression, especially in Turks and Moroccans.

## Figures and Tables

**Figure 1 ijerph-18-07134-f001:**
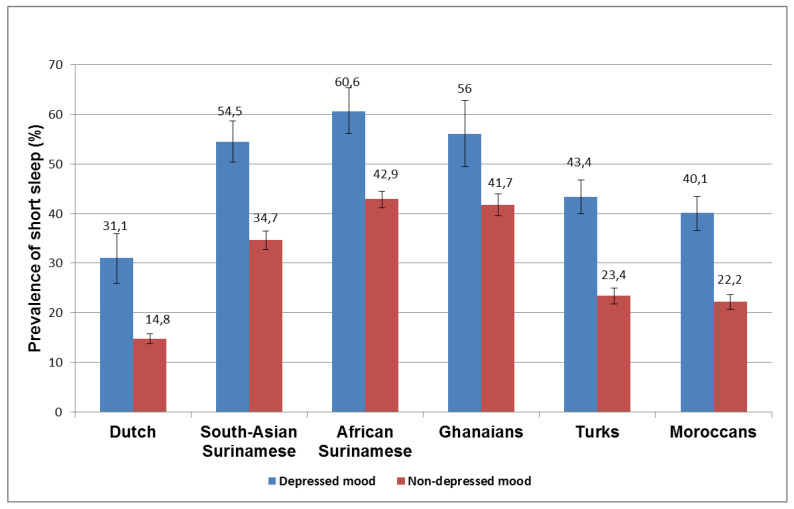
Association between depressed mood and short sleep duration among six ethnic groups living in Amsterdam.

**Figure 2 ijerph-18-07134-f002:**
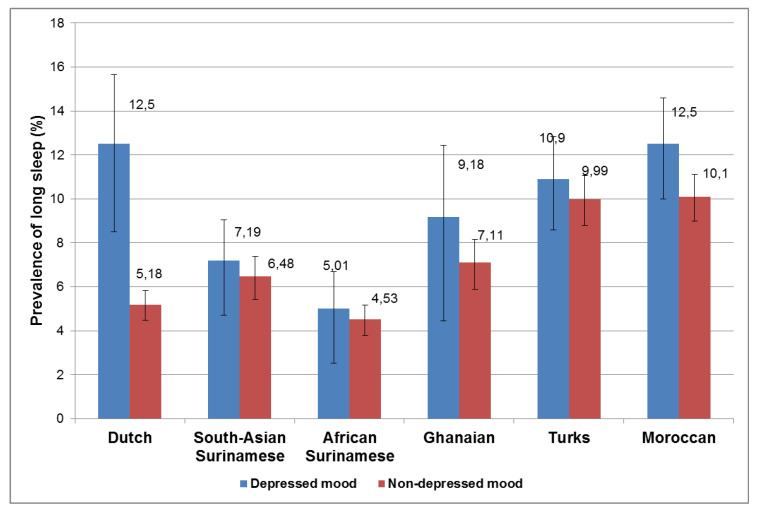
Association between depressed mood and long sleep duration among six ethnic groups living in Amsterdam.

**Table 1 ijerph-18-07134-t001:** Characteristics of the study population.

	Dutch	South-Asian Surinamese	African Surinamese	Ghanaians	Turks	Moroccans
	n = 4540	n = 2992	n = 4052	n = 2192	n = 3494	n = 3802
Age (years)	46.2 (45.7–46.6)	45.5 (45.0–46.0)	47.9 (47.6–48.3)	44.7 (44.3–45.2)	40.4 (39.9–40.8)	40.4 (40.0–40.8)
Men (%)	45.8 (44.4–47.3)	45.3 (43.5–47.1)	39.0 (37.5–40.5)	39.4 (37.3–41.4)	45.4 (43.7–47.0)	38.9 (37.4–40.5)
Sleep duration (hours)	7.3 (7.24, 7.29)	6.8 (6.79, 6.89)	6.6 (6.6, 6.7)	6.8 (6.71, 6.83)	7.1 (7.10, 7.19)	7.2 (7.12, 7.21)
Short sleep (% yes)	16.0 (14.9–17.1)	38.3 (36.6–40.1)	45.0 (43.5–46.5)	42.9 (40.9–45.0)	28.2 (26.8–29.7)	25.9 (24.5–27.3)
Long sleep (% yes)	5.7 (5.1–6.4)	6.61 (5.77–7.55)	4.60 (3.99–5.28)	7.25 (6.25–8.40)	10.2 (9.29–11.3)	10.7 (9.79–11.7)
Marital status						
Married (% yes)	37.8 (36.4–39.2)	34.4 (32.7–36.1)	18.4 (17.2–19.6)	18.2 (16.7–19.8)	61.1 (59.5–62.7)	58.5 (56.9–60.1)
Living together (%yes)	20.0 (18.9–21.2)	10.2 (9.16–11.3)	10.7 (9.84–11.7)	18.4 (16.8–20.0)	3.69 (3.12–4.37)	2.83 (2.35–3.41)
Single (% yes)	32.3 (30.9–33.7)	33.0 (31.3–34.7)	53.6 (52.1–55.2)	32.8 (30.9–34.8)	21.1 (19.8–22.5)	26.0 (24.6–27.4)
Widowed (% yes)	7.79 (7.04–8.60)	18.9 (17.6–20.4)	14.9 (13.8–16.0)	28.2 (26.4–30.1)	11.3 (10.3–12.3)	10.4 (9.47–11.4)
Divorced (% yes)	1.91 (1.55–2.36)	3.02 (2.45–3.69)	1.56 (1.23–1.99)	0.98 (0.64–1.48)	2.35 (1.92–2.93)	1.77 (1.39–2.24)
Depressed mood (%yes)	7.2 (6.5–8.0)	18.5 (17.1–19.9)	10.7 (9.79–11.7)	9.0 (7.9–10.2)	23.1 (21.7–24.5)	20.8 (19.5–22.1)
Educational level						
1st and 2nd category (%)	17.5 (16.4–18.6)	47.8 (45.9–49.5)	41.3 (39.7–42.8)	68.5 (66.5–70.3)	56.4 (54.8–58.0)	48.9 (47.3–50.5)
3rd category (%)	21.4 (20.6–23.1)	29.2 (27.6–30.9)	35.8 (34.3–37.2)	25.3 (23.5–27.1)	28.5 (27.1–30.0)	33.6 (32.1–35.1)
4th category (%)	60.7 (59.2–62.1)	22.9 (21.5–24.5)	22.9 (21.7–23.4)	6.27 (5.33–7.36)	15.0 (13.9–16.3)	17.5 (16.4–18.8)
Occupation						
Elementary (%)	1.78 (1.42–2.22)	10.8 (9.64–11.9)	6.97 (6.19–7.83)	63.6 (61.4–65.7)	19.7 (18.2–21.2)	17.9 (16.5–19.3)
Lower (%)	15.1 (14.1–16.2)	34.9 (33.2–36.8)	35.2 (33.7–36.7)	23.4 (21.6–25.4)	40.8 (38.9–42.7)	34.6 (32.9–36.4)
Middle (%)	23.4 (22.2–24.7)	31.1 (29.3–32.9)	35.5 (33.9–37.1)	9.02 (7.81–10.4)	24.6 (22.9–26.2)	28.9 (27.2–30.6)
Higher (%)	38.7 (37.2–40.1)	17.9 (16.6–19.5)	19.5 (18.2–20.8)	2.94 (2.27–3.79)	10.8 (9.70–12.0)	15.5 (14.2–16.9)
Scientific (%)	20.9 (19.8–22.2)	5.19 (4.41–6.10)	2.85 (2.36–3.44)	0.10 (0.64–1.56)	4.11 (3.42–4.93)	3.12 (2.53–3.83)
Current smoker (%yes)	24.7 (23.4–25.9)	28.4 (26.8–30.0)	31.4 (29.9–32.8)	4.54 (3.75–5.48)	34.2 (32.6–35.8)	13.5 (12.4–14.6)
Alcohol intake (%yes)	90.9 (90.1–91.8)	56.4 (54.6–58.2)	68.1 (66.7–69.5)	47.7 (46.1–50.2)	22.6 (21.3–24.0)	7.33 (6.55–8.20)
Psychotropic (% yes)	7.02 (6.37–7.87)	6.77 (5.93–7.73)	4.74 (4.13–5.44)	2.94 (2.31–3.72)	7.68 (6.85–8.60)	5.44 (4.76–6.20)

Data presented as means or percentages with 95% confidence interval (CI).

**Table 2 ijerph-18-07134-t002:** The association (expressed as odds ratio (OR)) of depressed mood with the prevalence of short sleep and with the prevalence of long sleep duration, by ethnicity.

	Dutch n = 4540	South-Asian Surinamese n = 2992	African Surinamese n = 4052	Ghanaians n = 2192	Turks n = 3494	Moroccans n = 3802
	OR (95%CI)	OR (95%CI)	OR (95%CI)	OR (95%CI)	OR (95%CI)	OR (95%CI)
Short sleep (<7 h/night)						
Model 1	2.8 (2.2–3.7)	2.3 (1.9–2.8)	2.3 (1.9–2.8)	1.9 (1.4–2.6)	2.5 (2.1–2.9)	2.4 (2.0–2.8)
Model 2	2.4 (1.9–3.2)	2.3 (1.9–2.8)	2.2 (1.8–2.8)	1.9 (1.4–2.6)	2.5 (2.1–2.9)	2.4 (2.0–2.8)
Model 3	2.5 (1.9–3.2)	2.2 (1.8–2.7)	2.2 (1.8–2.8)	1.9 (1.5–2.7)	2.4 (2.0–2.9)	2.3 (1.9–2.8)
PAF ^a^	6.7 (4.4–9.0)	8.9 (6.7–11.2)	4.7 (3.5–5.9)	3.5 (1.9–5.1)	15.5 (12.3–18.7)	13.9 (10.9–16.9)
Long sleep (≥9 h/night)						
Model 1	2.5 (1.8–3.6)	1.1 (0.7–1.5)	1.0 (0.7–1.7)	1.2 (0.7–2.1)	1.1 (0.9–1.5)	1.3 (1.0–1.6)
Model 2	2.3 (1.6–3.3)	0.9 (0.7–1.4)	0.9 (0.5–1.4)	1.2 (0.7–1.9)	1.0 (0.8–1.4)	1.2 (0.9–1.5)
Model 3	1.9 (1.3–2.8)	0.9 (0.6–1.3)	0.8 (0.5–1.4)	1.1 (0.7–1.9)	1.0 (0.8–1.3)	1.2 (0.8–1.4)
PAF ^a^	7.1 (2.1–11.7)	–	–	–	–	2.0 (−3.5, 7.3)

Model 1: age–sex adjusted; model 2: adjusted for model 1 plus marital status, education, and occupation; model 3: adjusted for model 2 plus smoking, alcohol consumption, and use of a psychotropic substance. PAF ^a^: Population attributable fraction.

## Data Availability

Data are available from the HELIUS study, a third party, Snijder, and Stronks who are affiliated with the HELIUS research cohort and are co-authors in this paper in accordance with HELIUS requirements for collaboration. Snijder is the Scientific Coordinator of HELIUS and may be contacted with further questions (heliuscoordinator@amsterdamumc.nl). Additionally, researchers interested in further collaboration with HELIUS may see the following URL: http://www.heliusstudy.nl/nl/researchers/collaboration (accessed on 29 June 2021).

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
