# Peer review of "Association between Depressed Mood and Sleep Duration among Various Ethnic Groups—The Helius Study"

_ijerph, 2021, doi:10.3390/ijerph18137134_

Round 1
Reviewer 1 Report
Thank you for opportunity to review the manuscript entitled "Association between Depressed Mood and Sleep Duration 2 among Various Ethnic Groups - The Helius Study" The work is well written and clarifies the results clearly. The great adventage is large and multi-etnical sample. Test is well applied. The conclusions are correct.
Few minor points:
1) Tables and figures are legible, however, in Figures 1 and 2 there is no p value and it is not clear whether these differences are statistically significant.
2) The authors assessed sleep duration using questionnaires and did not conduct additional studies (actigraphy or polysomnography). This is the limitation of this job.
3) Have the authors studied gender differences in different ethnic groups? This analysis would add value to the study.
Author Response
We thank the reviewer for the very important comments made. These are relevant contributions that will improve our manuscript. We have addressed the comments to the best of our ability and trust to have satisfied you on the raised concerns. Response to comments is attached.

Reviewer 2 Report
Ιn this cross sectional study the authors examined the possible association between depressed mood (DM) and sleep duration in a multi-ethnic population in Amsterdam, and the extent to which DM accounts for short and long sleep. The study was based on baseline data from the HELIUS (Healthy Life in an Urban Setting) study. The authors concluded that DM was significantly associated with short sleep in all ethnic groups after adjustment for other covariates. DM was not associated with long sleep except in Dutch. DM partly accounted for the prevalence of short sleep. It is an interesting study, however I would like to make some comments.
- It would be interesting to have some data according to gender as DM is more frequent in females. Additionally, there might be differences on education, occupation, smoking and alcohol intake according to gender.
- Do the authors have data on parenthood and number of children?
- Do the authors have data on the number of the members of the family that live together?
- Do the authors have data on other co-morbidities? Are there any data on BMI?
- Are there any data for naps during the day i.e. ‘siestas’?
- A limitation of the study is that the data were subjective, not measured objectively ex. actigraphy .
Author Response

(The authors gave the same response as above.)

Round 2
Reviewer 2 Report
The authors have replied to my comments adequately
Author Response
Reviewer comment
I appreciate the study results but I don't understand the premise that differences would be found among ethnic groups - why would that be? Genes? Socioeconomics? Discrimination? Noisy neighborhoods? Family size? Differential health care? Cultural tradition?
Response
We thank the reviewer/editor for the comments. The potential explanations for the observed differences between ethnic groups have been discussed in the manuscript e.g. the higher prevalence of depression in the Turkish and Moroccan population. More explanations were also highlighted in a previously published paper (Anujuo, K.; Stronks, K.; Snijder, M.B.; Jean-Louis, G.; Ogedegbe, G.; Agyemang, C. Ethnic differences in self-reported sleep duration in the Netherlands – the HELIUS Study. Sleep Med. 2015, 15, 1115-1121). For instance, psychosocial stress due to migration, perceived discrimination, congested and noisy residential environment, all negatively affect sleep duration and could vary among ethnic groups. We have now included these additional mechanisms in the discussion section of the revised manuscript. In addition, we have mentioned the lack of information on these mechanisms as a limitation of our study.
